# Optical Properties of Cylindrical Quantum Dots with Hyperbolic-Type Axial Potential under Applied Electric Field

**DOI:** 10.3390/nano12193367

**Published:** 2022-09-27

**Authors:** Esin Kasapoglu, Melike Behiye Yücel, Serpil Sakiroglu, Huseyin Sari, Carlos A. Duque

**Affiliations:** 1Department of Physics, Faculty of Science, Sivas Cumhuriyet University, 58140 Sivas, Turkey; 2Department of Physics, Faculty of Science, Akdeniz University, 07058 Antalya, Turkey; 3Physics Department, Faculty of Science, Dokuz Eylul University, 35390 Izmir, Turkey; 4Department of Mathematical and Natural Science Education, Faculty of Education, Sivas Cumhuriyet University, 58140 Sivas, Turkey; 5Grupo de Materia Condensada-UdeA, Instituto de Física, Facultad de Ciencias Exactas y Naturales, Universidad de Antioquia UdeA, Calle 70 No. 52-21, Medellín 050010, Colombia

**Keywords:** cylindrical quantum dot, optical absorption, hyperbolic potentials

## Abstract

In this paper, we have researched the electronic and optical properties of cylindrical quantum dot structures by selecting four different hyperbolic-type potentials in the axial direction under an axially-applied electric field. We have considered a position-dependent effective mass model in which both the smooth variation of the effective mass in the axial direction adjusted to the way the confining potentials change and its abrupt change in the radial direction have been considered in solving the eigenvalue differential equation. The calculations of the eigenvalue equation have been implemented considering both the Dirichlet conditions (zero flux) and the open boundary conditions (non-zero flux) in the planes perpendicular to the direction of the applied electric field, which guarantees the validity of the results presented in this study for quasi-steady states with extremely high lifetimes. We have used the diagonalization method combined with the finite element method to find the eigenvalues and eigenfunction of the confined electron in the cylindrical quantum dots. The numerical strategies that have been used for the solution of the differential equations allowed us to overcome the multiple problems that the boundary conditions present in the region of intersection of the flat and cylindrical faces that form the boundary of the heterostructure. To calculate the linear and third-order nonlinear optical absorption coefficients and relative changes in the refractive index, a two-level approach in the density matrix expansion is used. Our results show that the electronic and, therefore, optical properties of the structures focused on can be adjusted to obtain a suitable response for specific studies or goals by changing structural parameters such as the widths and depths of the potentials in the axial direction, as well as the electric field intensity.

## 1. Introduction

Quantum dots (QDs) are attractive structures that lead to interesting physical phenomena due to the restriction of the charge carrier movement in all spatial directions. By changing the size and shape of quantum dots, one can engineer and fine-tune the electronic and optical properties of such nanostructures [1,2,3,4]. Thanks to advances in modern fabrication techniques, QDs can be made into different shapes such as cubic [5,6], pyramidal [7], core/shell [8,9], spherical [10,11,12,13] and cylindrical [14,15,16,17,18,19,20]. Such structures present desirable optical features, which can be used to control and modulate the output intensity. The more the effects of the size and shape on the electronic and optical properties of QDs are important, the more the effects of the external perturbations such as magnetic, electric, intense laser fields and hydrostatic pressure are also important [14,15,16,17,18,19]. Applied external fields also affect the spectral features of QDs and so, any external field may also be used to control and tune the energy levels of electrons in a QD. It is clear that when manipulating the electronic energy spectrum of QDs, one may also adjust the absorption threshold frequency; doing so reflects the optical characteristics of the QD. In other words, one may control the absorption threshold frequency by technologically adjusting the QD growth. The advantage of cylindrical quantum dot (CQD) systems is to make it possible to control the energy spectrum with two structure parameters, such as the radius and height of the cylinder. This advantage gives more possibilities for investigation of the optical properties of nanostructures.

Due to the importance of size and geometric shape on the electronic and optical properties of low dimensional systems, CQDs with molecular-type axial potentials such as Woods–Saxon [21], Morse [22], Hulthen [23], Pöschl–Teller and Kratzer [24,25] have been studied extensively. A theoretical study of the effects of intense laser fields on the nonlinear properties of donor impurities in a spherical QD with Woods–Saxon potential was performed within the matrix diagonalization method and in the effective mass approximation [21]. The authors considered several configurations of the barrier height, the dot radius, the barrier slope of the confinement potential and the incident intense laser radiation, finding that all these factors can influence the nonlinear optical properties strongly. Within the framework of perturbation theory and variational method, Hayrapetyan et al. [22] calculated the electronic states and optical properties of CQDs with Morse confining potential made of GaAs in the presence of parallel electrical and magnetic fields. A generalized Hulthen potential has been used in the study of the electron-related linear and nonlinear optical properties in spherical QDs by Onyeaju and co-workers [23]. The advantage of the proposed potential and the approximations considered by the authors is that it is possible to obtain an analytical solution for the eigenvalue differential equation in terms of hypergeometric-type functions. Among the multiple advantages of the study of spherical systems is the simplicity for the solution of differential equations, particularly in the absence of electrostatic interactions. In general, spherical systems involve only the solution of a one-dimensional differential equation associated with the radial coordinate. Spherical symmetry also facilitates boundary conditions in regions of interfaces between well and barrier materials.

In the case of the CQDs reported in Refs. [14,15,16,17,18,19,20], the following observations and comments should be taken into account. In most cases, researchers consider an infinite confinement potential either in the axial direction or in the radial direction. This allows the investigator to decouple the eigenvalue differential equation into two parts, one radial and one axial; in this case there is no need to establish mathematical approximations to solve the eigenvalues problem. This infinite potential approximation is good when the physical system corresponds, for example, to a CQD made of a semiconductor that is surrounded by a vacuum or to a CQD made of a semiconductor surrounded by another material whose energy gap is significantly higher. The same kind of separation of differential equations for the radial and axial parts has been established by most researchers who have considered CQD with finite confinement potentials. In this case, the approximation is more or less good when the QD dimensions are large enough in such a way that the electron wave functions are essentially confined to the QD region. The approximation collapses and leads to dramatically relevant errors when the QD dimensions are in the same order of magnitude or less than the effective Bohr radius of the dot material.

To avoid the boundary problems associated with the region where the cylindrical wall connects with the two planes in a CQD, most researchers resort to using an infinite confining potential on either the cylindrical wall or the two plane walls that are perpendicular to the axial axis. This, again, taking into account the azimuthal symmetry of the problem, allows us to separate the 2D differential equation (which has radial and axial parts) into two decoupled differential equations, again finding simple solutions to the eigenvalue problem [14,15,16,17,18,19,20]. This is a much less realistic problem than the one where the confining potential is infinite in all spatial directions of the QD boundary. To address the complete problem of a QD surrounded by a material that provides a finite potential barrier, it is then necessary to resort to methods that allow solving the eigenvalue equations considering the correct form of the boundary conditions. In general, it is possible to resort to numerical methods where the differential operators are discretized in the space both inside and outside the QD region. Among the useful numerical methods, we can mention the finite difference method and the finite element method (FEM). In those cases where there are singularities associated with electrostatic potentials, for example, mesh refinement can be used to be able to adequately account for the particularities of the problem. These methods simplify the inclusion of effects such as spatial variation of the effective mass and dielectric constants that are associated with different semiconductor materials for both the dot and barrier regions.

In this context, taking into account all the problems mentioned above concerning the dot-barrier boundary conditions and the spatial variations of effective mass and dielectric constant, in this article we have decided to study the combined effects of an externally applied electric field as well as the size and geometric shape on the electronic and optical properties of the CQD ensemble, formed by the selection of some hyperbolic-type potentials [26,27] in the axial direction. Initially, we will consider an infinite axial potential combined with a finite radial potential but, in general, the article will be devoted to the consideration of finite confining potentials in both the radial and axial directions. In that sense, we will resort to the FEM to solve the eigenvalue differential equations. Using the advantages of the FEM, we will take into account the effects of spatial variation in the effective mass. Furthermore, we will present a theoretical analysis of the inter-subband absorption coefficients including contributions of linear and nonlinear terms for the electron confined within these potentials. The paper is organized as follows: Section 2 contains the theoretical description, the obtained results are discussed in Section 3 and, finally, the conclusions are given in Section 4.

## 2. Theoretical Model

We consider a CQD that has different hyperbolic-type axial potentials combined with finite step-like radial potential under the effects of an axial applied electric field. Considering a position dependent effective mass and the conduction band effective mass approximation, the Hamiltonian in the cylindrical coordinates for the system under the electric field is as follows [15,16,17,18,28,29,30]
(1)$$H=-\frac{\hbar^{2}}{2}\;\left(\frac{\partial }{\partial \rho }\;{\hat{u}}_{\rho }+\frac{1}{\rho }\;\frac{\partial }{\partial \theta }\;{\hat{u}}_{\theta }+\frac{\partial }{\partial z}\;{\hat{u}}_{z}\right)\cdot \left(\frac{1}{m^{*}\left(\vec{r}\right)}\right)\left(\frac{\partial }{\partial \rho }\;{\hat{u}}_{\rho }+\frac{1}{\rho }\;\frac{\partial }{\partial \theta }\;{\hat{u}}_{\theta }+\frac{\partial }{\partial z}\;{\hat{u}}_{z}\right)+V\left(\vec{r}\right)+\left|e\right|\;F\;z\;.$$

The corresponding Schrödinger equation is given by
(2)Hψ(ρ,θ,z)=Eψ(ρ,θ,z).

In Equation (Equation 1), *e* is the electron charge and *F* is the strength of the electric field, which is applied parallel to the growth direction (*z*-axis). Additionally, $m^{*}\left(\vec{r}\right)$ and $V(\vec{r})$ are the position-dependent electron effective mass and confinement potential, respectively. In this study, we deal with a cylindrical heterostructure where the axial symmetry is preserved both for the effective mass and confinement potential. Consequently, the wave function in Equation (Equation 2) can be written as ψ(ρ,θ,z)=φ(ρ,z)exp(ilθ), where *l* is an integer number. Under such condition, Equation (Equation 2) reduces to [31]
(3)$$\begin{array}{c} \;\left\{-\frac{\hbar^{2}}{2}\;\left(\frac{\partial }{\partial r}\;{\hat{u}}_{r}+\frac{\partial }{\partial z}\;{\hat{u}}_{z}\right)\cdot \left(\frac{1}{m^{*}\left(r,z\right)}\right)\left(\frac{\partial }{\partial r}\;{\hat{u}}_{r}+\frac{\partial }{\partial z}\;{\hat{u}}_{z}\right)+\frac{\hbar^{2}\;l^{2}}{2\;m^{*}\left(r,z\right)}+V\left(\vec{r}\right)+\left|e\right|\;F\;z\right\}\;\varphi \left(\rho ,z\right)\\ \;\;\;=E\;\varphi (\rho ,z)\;. \end{array}$$

The functional forms of the confinement potential and electron effective mass are given by
(4)$$V\left(\rho ,z\right)=\left\{\begin{array}{cc} V_{n}\left(z\right), & \rho \leq R\;,\\ V_{0}, & \rho >R\;, \end{array}\right.$$
and
(5)$$m^{*}\left(\rho ,z\right)=\left\{\begin{array}{cc} m_{n}^{*}\left(z\right), & \rho \leq R\;,\\ m_{Al_{x}Ga_{1-x}As}^{*}, & \rho >R\;, \end{array}\right.$$
where
(6)$$V_{n}\left(z\right)=A\;\frac{\sinh^{r}\left(\frac{z}{k}\right)}{\cosh^{s}\left(\frac{z}{k}\right)}+B\;.$$

Here, *R* is dot radius and *k* is related to the dot length. $V_{0}$ is the depth of the confinement potential between the dot and barrier regions. The constants *A*, *B*, *r* and *s* are chosen in order to provide several axial confinement potentials. In this work we deal with the following axial potentials, which can give rise to finite or infinite confinements and single or double structures:(7)$$V_{1}\left(z\right)=V_{0}\;\sinh^{2}\left(\frac{z}{k}\right)\;,$$
(8)$$V_{2}\left(z\right)=\left\{\begin{array}{cc} V_{0}\;\sinh^{2}\left(\frac{z}{k}\right), & \mathrm{if}\;\;|z|\leq 0.8813\;k\;,\\ V_{0}, & \mathrm{elsewhere}\;, \end{array}\right.$$
(9)$$V_{3}\left(z\right)=-V_{0}\;\frac{1}{\cosh^{2}\left(\frac{z}{k}\right)}+V_{0}\;,$$
and
(10)$$V_{4}\left(z\right)=-4\;V_{0}\;\frac{\sinh^{2}\left(\frac{z}{k}\right)}{\cosh^{4}\left(\frac{z}{k}\right)}+V_{0}\;.$$

For example, the potential $V_{1}\left(z\right)$ ($V_{2}\left(z\right)$) represents an infinite (finite) single QW potential. Furthermore, $V_{3}\left(z\right)$ corresponds to the modified Pöschl–Teller potential for (r,s)=(0,2), and $V_{4}\left(z\right)$ represents a double QD for (r,s)=(2,4). Note that all the potentials have been shifted in such a way that the bottom of the potential corresponds to the zero of the energy.

Considering that the dependence of the potential barrier $V_{n}$ on the aluminum concentration *x* is given by $V_{n}=a\;\left(b\;x+c\;x^{2}\right)$, in order to find the *z*-dependence of the effective mass in Equation (Equation 5), $m_{n}^{*}\left(z\right)$, we use the following expression to connect the spatial variation of the aluminum concentration and the potentials given by Equations (7)–(10):(11)$$x\left(z\right)=\frac{-b+\sqrt{b^{2}+\frac{4\;c\;V_{n}}{a}}}{2\;c}\;,$$
where $V_{n}\left(z\right)$ in Equations (7)–(10) are given in meV with a=0.658, b=1155 meV and c=370 meV [5]. Once the *z*-dependence of the aluminum concentration is obtained, the effective mass for the electron in Equation (Equation 5) is given by
(12)$$m_{n}^{*}\left(z\right)=m_{GaAs}+\left(m_{AlAs}-m_{GaAs}\right)\;x\left(z\right)\;,$$
where $m_{GaAs}=0.067\;m_{0}$ and $m_{AlAs}=0.124\;m_{0}$ ($m_{0}$ is the free electron mass) [5].

The local droplet epitaxy (LDE) technique allows us to obtain structures of GaAs surrounded by Al_x_Ga_1−x_As with axial symmetry with zero or almost zero strain effects; see, for instance, Stemmann et al. [32], and references therein. Once the nano-hole is obtained, a growth process of alternate layers of Al_x_Ga_1−x_As can be started by controlling the value of the *x*-concentration of aluminum in such a way that a desired potential profile can be achieved. In this sense, we want to emphasize that the potentials that we have modeled in this article via Equations (7)–(10) are perfectly plausible from the experimental point of view.

Due to the form presented by Equation (Equation 4), it can clearly be seen that Equation (Equation 3) is not separable. That means that, in general, from Equation (Equation 3), it is not possible to obtain two independent differential equations for the ρ- and *z*-coordinates. In the case of Equation (Equation 7), where it can be taken into account that the potential along the *z*-direction diverges, it is not very wrong to consider the approximation V(ρ,z)∼$V_{\rho }\left(\rho \right)+V_{1}\left(z\right)$, with $V_{\rho }=0$ for ρ≤R and $V_{\rho }=V_{0}$ for ρ>R. Thus, in particular, we can write $\varphi_{1}\left(\rho ,z\right)=N\;h_{1}\left(\rho \right)\;h_{2}\left(z\right)$, where *N* is the normalization constant. In this case, assuming that the effective mass is constant throughout the structure (it can be considered that the mass throughout the structure corresponds to the value of the QD region, $m_{GaAs}^{*}$), we obtain from Equation (Equation 3) the following two differential equations:(13)$$\left[-\frac{\hbar^{2}}{2\;m_{GaAs}^{*}}\;\left(\frac{\partial^{2}}{\partial \rho^{2}}+\frac{1}{\rho }\;\frac{\partial }{\partial \rho }\right)+\frac{\hbar^{2}\;l^{2}}{2\;m_{GaAs}^{*}}+V_{\rho }\left(\rho \right)\right]\;h_{1}\left(\rho \right)=E_{\rho }\;h_{1}\left(\rho \right)$$
and
(14)$$\left[-\frac{\hbar^{2}}{2\;m_{GaAs}^{*}}\;\frac{\partial^{2}}{\partial z^{2}}+V_{1}\left(z\right)+\left|e\right|\;F\;z\right]\;h_{2}\left(z\right)=E_{z}\;h_{2}\left(z\right)\;.$$

Considering l=0 (corresponding to the ground state), the ground state eigenfunction associated to the Equation (Equation 13) is given by [33,34]
(15)$$h_{1}\left(\rho \right)=\left\{\begin{array}{cc} J_{0}\left(k_{0}\;\rho \right), & \rho \leq R\;,\\ \frac{J_{0}\left(k_{0}\;R\right)}{K_{0}\left(q_{0}\;R\right)}K_{0}\left(q_{0}\;\rho \right), & \rho >R\;, \end{array}\right.$$
where $J_{0}$ and $K_{0}$ are the Bessel functions of the first kind and modified Bessel functions, respectively. Here, $k_{0}={\left(\frac{2\;m_{GaAs}^{*}}{\hbar^{2}}\;E_{\rho }^{1}\right)}^{1/2}$ and $q_{0}={\left[\frac{2\;m_{GaAs}^{*}}{\hbar^{2}}\;\left(V_{0}-E_{\rho }^{1}\right)\right]}^{1/2}$. The energy associated to the radial confinement, $E_{\rho }^{1}$, is obtained from the continuity condition at the point of r=R of the $h_{1}$-wave function and its first derivative. To solve the Equation (Equation 13), we used the diagonalization method by choosing orthonormal base functions that are solutions of an infinite square QW with *L*-width [35,36,37]. The value of *L* is determined according to the convergence of the energy eigenvalues. Once the energies associated with the ground state have been obtained for $h_{1}$ in Equation (Equation 15) ($E_{\rho }^{1}$) and the different solutions for the lower energies in Equation (Equation 14) ($E_{z}^{n}$, n=1,2,3,…), the energy of the lowest states with l=0 corresponding to $\varphi_{1}$ are given by $E_{1}^{n}=E_{\rho }^{1}+E_{z}^{n}$ with n=1,2,3,….

Taking into account that, in general, the potentials in Equation (Equation 3) cannot be written as a sum of two independent terms associated with the radial and axial coordinates, the solution of Equation (Equation 3) necessarily involves a process of solving a two-dimensional eigenvalue differential equation. In our case, we have decided to implement a numerical calculation using the finite element method (FEM). In this case, we resort to the COMSOL-Multiphysics licensed software and particularly to the two-dimensional axi-symmetric module [38,39,40,41,42,43,44].

Figure 1a shows the scheme of the structure used to solve the two-dimensional eigenvalue differential equation, Equation (Equation 3), using the finite element method. Figure 1b shows the mesh used with its refinement in the QD region to implement the finite element method. The 3D cylindrical structure is obtained by rotating the rectangular region around the *z*-axis. The Dirichlet conditions are imposed on the sides AB, BC and CD corresponding to a rectangle of dimensions $L_{1}=40$ nm and $L_{2}=60$ nm, which is large enough to guarantee the convergence of at least 15 states for each value of l=0,±1,±2,±3. A fine mesh with double refinement in the QD region is considered. The parameters for the mesh used in this study in the calculation of FEM are: 3135 triangles, 201 edge elements, 8 vertex elements, 0.4333 for the minimum quality of elements, 0.899 for the average quality of elements, 0.1121 for the ratio of element area and a mesh area of 2400 nm^2^.

Now that the eigenvalues and eigenfunctions of the electron in the quantum dot are known, we can proceed to calculate the optical properties of the CQD with different hyperbolic-type axial potentials. By employing density matrix formalism and the perturbation expansion method, the first order linear, the third order nonlinear and the total absorption coefficient (AC) terms for transitions between any two energy levels are found as follows [45,46,47,48,49,50], respectively:(16)$$\beta_{1\to f}^{\left(1\right)}\left(\omega \right)=\omega \;\sqrt{\frac{\mu_{0}}{\varepsilon_{r}\;\varepsilon_{0}}}\;\;\frac{|M_{1f}{|}^{2}\;\sigma_{v}\;\hbar \;\Gamma_{1f}}{{\left(E_{1f}-\hbar \;\omega \right)}^{2}+{\left(\hbar \;\Gamma_{1f}\right)}^{2}}\;,$$
(17)$$\begin{array}{ccc} \;\beta_{1\to f}^{\left(3\right)}\left(\omega ,I\right) & = & -\omega \;\sqrt{\frac{\mu_{0}}{\varepsilon_{r}\;\varepsilon_{0}}}\;\left(\frac{I}{2\;n_{r}\;\varepsilon_{0}\;c}\right)\;\;\frac{|M_{1f}{|}^{2}\;\sigma_{v}\;\hbar \;\Gamma_{1f}}{{\left[{\left(E_{1f}-\hbar \;\omega \right)}^{2}+{\left(\hbar \;\Gamma_{1f}\right)}^{2}\right]}^{2}}\\ \; & \; & \times \left[4\;|M_{1f}{|}^{2}-\frac{|M_{ff}-M_{11}{|}^{2}\;\left[3\;E_{1f}^{2}-4\;E_{1f}\;\hbar \;\omega +\hbar^{2}\left(\omega^{2}-\Gamma_{1f}^{2}\right)\right]}{E_{1f}^{2}+{\left(\hbar \;\Gamma_{1f}\right)}^{2}}\right]\;, \end{array}$$
(18)$$\beta \left(\omega ,I\right)=\sum_{f=2}^{19}\left(\beta_{1\to f}^{\left(1\right)}\left(\omega \right)+\beta_{1\to f}^{\left(3\right)}\left(\omega ,I\right)\right)\;,$$
where $\sigma_{v}$ is the carrier density in the system, $\mu_{0}$ and $\varepsilon_{0}$ are the vacuum permeability and permittivity, respectively, $\varepsilon_{r}$ is the GaAs QD permittivity, $n_{r}=\sqrt{\varepsilon_{r}}$ is the GaAs QD refraction index, $E_{1f}=E_{f}-E_{1}$ is the energy difference between first three electronic states, $M_{i\;j}=\langle \psi_{i}\left|e\;z\right|\psi_{j}\rangle$ is dipole moment matrix element (note that for discussion purposes, the $M_{i\;j}=M_{i\;j}/e$ parameter will be used), $\Gamma_{1f}$ is the relaxation rate which equals to the inverse relaxation time $T_{1f}$, *c* is the speed of the light in free space and *I* is the incident optical intensity, which is defined as $I=\frac{2\;n_{r}}{\mu_{0}\;c}\;{\left|E\left(\omega \right)\right|}^{2}$. In the case of relative changes of the refraction index coefficient, the corresponding expressions are
(19)$$\frac{\Delta n_{1\to f}^{\left(1\right)}\left(\omega \right)}{n_{r}}=\frac{\sigma_{v}\;{\left|M_{1f}\right|}^{2}}{2\;\varepsilon_{0}\;n_{r}^{2}}\;\;\frac{E_{1f}-\hbar \;\omega }{{\left(E_{1f}-\hbar \;\omega \right)}^{2}+{\left(\hbar \;\Gamma_{1f}\right)}^{2}}\;,$$
(20)$$\begin{array}{c} \;\frac{\Delta n_{1\to f}^{\left(3\right)}\left(\omega ,I\right)}{n_{r}}=-\frac{\mu_{0}\;c\;I\;\sigma_{v}\;{\left|M_{1f}\right|}^{2}}{4\;\varepsilon_{0}\;n_{r}^{3}}\;\;\frac{E_{1f}-\hbar \;\omega }{{\left[{\left(E_{1f}-\hbar \;\omega \right)}^{2}+{\left(\hbar \;\Gamma_{1f}\right)}^{2}\right]}^{2}}\\ \times \left[4\;|M_{1f}{|}^{2}-\frac{|M_{ff}-M_{11}{|}^{2}}{E_{1f}^{2}+{\left(\hbar \;\Gamma_{1f}\right)}^{2}}\;\;\left\{E_{1f}\;\left(E_{1f}-\hbar \;\omega \right)-{\left(\hbar \;\Gamma_{1f}\right)}^{2}-{\left(\hbar \;\Gamma_{1f}\right)}^{2}\frac{\left(2\;E_{1f}-\hbar \;\omega \right)}{\left(E_{1f}-\hbar \;\omega \right)}\right\}\right]\;, \end{array}$$
and
(21)$$\frac{\Delta n(\omega ,I)}{n_{r}}=\sum_{f=2}^{19}\left(\frac{\Delta n_{1\to f}^{\left(1\right)}\left(\omega \right)}{n_{r}}+\frac{\Delta n_{1\to f}^{\left(3\right)}\left(\omega ,I\right)}{n_{r}}\right)\;.$$

## 3. Results and Discussion

In our calculations, the following parameters have been used: ε=13.18, $n_{r}=3.6$, $T_{1f}=1.0$ ps, $\Gamma_{1f}=1/T_{1f}$, $\mu_{0}=4\pi \times 10^{-7}$ H/m, $\varepsilon_{0}=8.85\times 10^{-12}$ F m ${}^{-1}$, $I=1.5\times 10^{8}$ W m ${}^{-2}$ and $\sigma_{v}=3.0\times 10^{22}$ m ${}^{-3}$. Additionally, $V_{0}=0.658\;\left(1155\;x+370\;x^{2}\right)$ meV, with x=0.3.

In Figure 2, our findings are reported for the lowest energies for a confined electron within the CQD as a function of the simultaneous variation of the *k*-width in the axial direction and the *R*-radius in the radial direction (note that k=R). Calculations are for zero applied electric field and using the $V_{1}\left(z\right)$-model for the axial infinite confinement potential (see Equation (Equation 7)). In Figure 2a,b, the solid lines correspond to the results obtained by the solution of the full two-variable V(ρ,z) confinement potential. In Figure 2a, the five lowest solutions are reported for each l=0,±1,±2 value. In Figure 2b, the lowest five states are reported for l=0. The full symbols in Figure 2b are obtained by the approximation $V\left(\rho ,z\right)=V_{\rho }\left(\rho \right)+V_{1}\left(z\right)$, using only the first solution for the radial differential equation. In Figure 2b, the (i,j) labels identify the number of wave function antinodes, both in the radial (i) and axial (j) directions. In this case, we have used the same value of the GaAs effective mass in both the dot and barrier regions. Note that due to the dot symmetry, in Figure 2a, the states with *l*-positive and *l*-negative are degenerate. As *k* and/or *R* increases, the decreasing character of all of the energies taken into consideration is the result of the significant increase in the QD’s volume due to the weak confinement. As the dimensions of the QD decrease, the number of confined states becomes smaller. For k=10 nm, there are six confined states, whereas for k=5 nm, there are two confined states, with one of them very close to the limit of the finite radial potential barrier. The appearance of crossovers between energy states at certain values of the *k* and/or *R* parameters is based on the changes in the volume dominated by the increase in the radial direction of QD. In Figure 2b, we have selected the states with l=0 for our axis-symmetric model; see Equation (Equation 3). The solid dots in Figure 2b correspond to the calculation done through Equations (13) and (14) with the condition l=0 that gives rise to the radial solution in Equation (Equation 15). For the transcendental equation that appears from involving the boundary conditions in Equation (Equation 15), only the first solution has been used. Therefore, the different energy levels shown with solid symbols come from considering the different solutions of Equation (Equation 13) combined with the first solution associated with Equation (Equation 15). It is evident that the solutions of both models (solid lines and full symbols) coincide with differences of less than 0.001 meV. This is justified by the fact that the axial potential of the $V_{1}\left(z\right)$ model (see Equation (Equation 7)) diverges to infinity as *k* increases. This is what allows us to write the total potential as a sum of two independent potentials and allows the total wave function to be written as the product of two independent functions in the radial and axial directions. It is clear from Figure 2b that regardless of the values of k=R, the ground state always has one antinode in the radial and axial directions. For k=R=10 nm, the first excited state with l=0 has one antinode along the radial direction and two along the axial direction, while for k=R=20 nm, the first excited state with l=0 has two antinodes along the radial direction and one along the axial direction.

Considering that, in this work, the incident radiation used to generate the transitions between confined states has *z* polarization, that is, along the axial axis (see the dipole matrix element defined after Equation (Equation 18)), it is evident that when considering transitions from the ground state (which have only one antinode along the radial and axial directions), the only ones allowed (such as $M_{i\;j}\neq 0$) are those where the final state also has the same number of antinodes along each of the two *x* and *y* directions of space that give rise to radial confinement. In that sense, it is clear that for the calculation of the optical properties using Equations (16)–(21), only transitions between states with l=0 should be considered, both for the initial and final states.

For the CQD under consideration in the absence and presence of the electric field, the variation of the axial confinement potentials given by Equations (8)–(10) and the wave functions related to the first four electronic levels versus the *z*-coordinate are given in Figure 3a–f. In the absence of the electric field, all axial potentials are symmetric, while the applied electric field causes the bending of the structure and break down of the symmetry. As seen in Figure 3c, the $V_{4}\left(z\right)$ axial finite potential of the CQD represents the double QW, while the other axial finite potentials represent the hyperbolic single QWs. Particularly, the potential shown in Figure 3a behaves as a nearly parabolic model of finite height. In Figure 3c, when there is no electric field, the ground state is two-folded degenerate and the probability of finding of the electron in both wells in the axial direction is the same, whereas the axially applied electric field causes the bending of the well, leading to a disappearance of the degeneracy and electrons locating mainly in the left well. Furthermore, as seen in this figure, the electric field effect for the well with the $V_{4}\left(z\right)$, which has a larger effective length, is stronger compared to the single QWs in the axial direction. In Figure 3d, the bottom of the quasi-parabolic potential well is located at z=−2.2 nm and has an energy of −5.58 meV. Clearly, the electric field effect on this type of quasi-parabolic potential is responsible for shifting the bottom of the potential well without essentially changing the shape of the spatially variable potential, mainly in the dot region. This means that the energy levels are displaced by the effect of the electric field, but the energy distance between them remains constant. This will later be seen as a quasi-overlap of the resonant peaks in the optical properties under the presence or absence of an electric field, particularly for the case of the finite potential $V_{2}\left(z\right)$. In our calculation model, we have used a cylindrical region of height $L_{2}=60$ nm where we apply the Dirichlet boundary conditions (see Figure 1a). This allows us to guarantee that all the solutions of the eigenvalue differential equation in the presence of electric field, applied along the axial direction, correspond to stationary states whose energies are real numbers. From the general solution of the differential equation, confined states associated with the outside region of the structure will appear, that is, within the potential barrier region, which comes from the infinite confinement that is imposed on the outside border shown in Figure 1b, a situation that will be seen later in our further results.

We have repeated the calculations considering open boundaries at $z=\pm L_{2}/2$. In this case, the number of solutions of the Schrödinger equation is significantly higher for energies below the height of the potential barrier. It is important to say that now all the obtained eigenvalues are complex numbers where the real part corresponds to the energy of the state and the imaginary part provides information about the corresponding state lifetime. In this second procedure in the set of solutions, we find the same wavefunctions (solutions) reported in Figure 3e–f, but with very low oscillations towards the left region of the structure. The imaginary parts of the eigenvalues are in this case very small. Considering that the lifetime of the states is proportional to the inverse of the imaginary part of the eigenvalue, we have obtained that, again considering open boundary conditions, the states are quasi-stationary with a very long lifetime. This leads us to conclude the validity of the results presented in Figure 3e–f and to affirm that the optical properties calculated in this article (see the following) are correct.

Figure 4a–f show the six lowest energy values of the electron confined within the CQD with respect to the order of axial potentials in the Figure 3a–f as a function of the *k*-well width in the axial direction and *R*-radius in the radial direction. Figure 5a–f shows the corresponding energy differences between the ground state and the first five excited states (note that the *k* and *R* parameters are taken as equal). Results in the absence (upper panels) and presence (lower panels) of the electric field are for $V_{2}\left(z\right)$, $V_{3}\left(z\right)$ and $V_{4}\left(z\right)$, respectively. For all axial finite confinement potentials, it is observed that the energies taken into consideration are reduced by increasing the dot sizes due to the weakening of the geometric confinement. Usually, the electric field also causes the energies to decrease. In large wells, energy levels begin to become more sensitive to the electric field. Note, for example, that in Figure 4d,e, the variations presented by the ground state in the range of calculated dimensions are 120 meV and 115 meV, while in the case of Figure 4e, this variation is 184 meV. It should be noted that the change of the energies in Figure 4a,b,d,e corresponding to the potentials $V_{2}\left(z\right)$ and $V_{3}\left(z\right)$ versus the CQD dimensions is almost the same for the cases both with and without electric field. Comparing Figure 4a,d, in the first, the variation of the ground state in the calculated range is 112 meV, while in the second, it is 120 meV with a difference between the two of 8 meV. In the case of Figure 4b,e, the respective changes are 100 meV and 115 meV, with a difference of 5 meV. Comparing Figure 4c,f the obtained values are 124 meV and 184 meV, with a variation between them of 60 meV.

The dependence of the energy difference on the dot sizes in Figure 5a,d is exactly the same. For this analysis, only the decreasing character of the transition energies should be taken into account as the dimensions k=R of the QD increase. Note, for example, that in Figure 5a, the transition energy $E_{16}$ increases as k=R increases from 5 nm to 11 nm and then decreases. The increasing part of the transition energy is due to the fact that, in that part of the process, the one that corresponds to the ground state confined to the QD participates as the initial state; this state always decreases with the dimensions of the QD. The process is carried out towards a final state that, in said range of dimensions of the structure, always corresponds to an unbound state of the QD, but is confined to the cylindrical region of dimensions $\pi \;L_{1}^{2}\;L_{2}$; see Figure 1. The same occurs for the other lower order transitions that are increasing in energy, while the final state corresponds to states not bounded to the QD. The electric field effect on energies corresponding to double QW potential given by $V_{4}\left(z\right)$ is more dominant in contrast to single QWs due to the large effective well width. Except for the $V_{4}\left(z\right)$ double well potential, the difference between the related energy levels for the other two single well potentials, $V_{2}\left(z\right)$ and $V_{3}\left(z\right)$, is a decreasing function of the dot size in the presence of the electric field. This is already expected, since the electric field generally causes the electronic states to shift to lower energies. The electric field effect on double-well potential, in Figure 5c,f, is different with respect to the single well potentials. (i) In the absence of the electric field, $E_{12}$ decreases down to k=10 nm, and will later become zero due to the two-folded degenerate of ground state energy. (ii) In the presence of the electric field, the degeneracy disappears and $E_{12}$ firstly decreases; then, it turns on $E_{13}$, which increases with the dot dimensions. As is known, applying an external electric field causes distortion of the symmetries of the heterostructures or, conversely, in an asymmetric heterostructure, symmetry can be regained. In this sense, the electric field can be used to increase the magnitude of the electric dipole moments associated with optical transitions and cause red shifts in the resonant structures of optical spectra, as electron states generally shift to lower energies.

In Figure 6a–f, we present the reduced square of the dipole matrix element between the lowest confined electron states ($M_{1n}^{2}/e^{2}$, n=2,3,4,5,6) within the CQD as a function of the simultaneous variation of the *k*-width in the axial direction and the *R*-radius in the radial direction. Calculations are for zero and 50 kV/cm applied electric field with l=0. The results are reported for the three considered finite axial confinement potentials as in previous figures. It is observed from the different panels that there will be essentially only one allowed transition and that, in general, this visible transition is made up of the mixture of transitions between different states depending on the value of the k=R parameter. This effect is the result of the crossing between states in Figure 4, where an exchange of symmetries of the wave functions is presented. The systematic increase in dipole matrix elements with increasing dimensions of the QD is associated with a greater spatial extension of the wave functions within the QD. The oscillatory character of $M_{16}^{2}$ in the range of k=R>15 nm in Figure 6c is compensated for by the appearance of the term $M_{15}^{2}$. Between both transitions, the total value of the dipole matrix elements responsible for the optical properties is compensated. When comparing the different results, it can be seen that the system that is most sensitive to the effects of electric fields is that of the potential $V_{4}\left(z\right)$, which is consistent with a greater spatial extension of the structure along the axial direction and which allows the electric field effects to be strengthened.

To see the effect of structure dimensions, geometric shape and electric field on the total optical absorption coefficient (AC) and on the total relative changes of the refraction index coefficient (RIC) of CQD with different axial potentials, we present the optical coefficients in Figure 7a–f. Solid lines are for zero electric field, whereas dotted lines are for F=50 kV/cm. We choose dimensions of the structures such that the existence of at least two states confined to the interior of the QD is guaranteed. We can explain our results considering the well-known properties of the optical properties such as: (i) the AC peak positions (or the zero value of the RIC) shift towards blue (red) due to the increasing (decreasing) of the energy difference between the respective energy levels; (ii) the peak amplitude of ACs and RICs increase due mainly to the increase in dipole matrix element; and (iii) in the case of the AC, the transition energy between participating states is also relevant, since it directly affects the magnitude of the resonant structure. The results show the following characteristics. In general, all the optical coefficients present a redshift as the value of the *k* and/or *R* parameter increases, which is consistent with a decrease in the transition energy as the volume of the structure increases. The structure that is less sensitive to the effects of the electric field, in fact, almost imperceptibly, is the one corresponding to the potential $V_{2}\left(z\right)$; see Figure 7a,d. Despite being a finite potential, clearly, the system presents a quasi-parabolic potential within the structure. Note from Figure 3a,d that the presence of the electric field implies a red shift of the energies, but the energy difference between levels, or transition energy, remains essentially constant. The increasing behavior of the magnitude of the different structures in the RICs as the dimensions of the QD increase is in accordance with the increase of the dipole matrix elements in Figure 6a–f. The appearance of two structures in the ACs of Figure 7a,b in the regimes of large k=R shows that the third-order nonlinear term acquires increasingly important values in these systems. Additionally, it can be seen in the same two figures that, although the dipole matrix element grows with structure dimensions, the resonant structures decrease due to the redshift of the spectra or the diminishing of transition energies. It is clear that the resonant structures in ACs depend simultaneously on the transition energy and on the dipole matrix elements. In general, far from the RIC zero, the optical coefficient tends asymptotically to zero, as shown by the results in Figure 7d,e. However, that is not the case in Figure 7f. This is explained by the fact that the transition between quasi-degenerate states has a significantly important value, as shown by the insets in Figure 7c,f. Finally, as stated before, the electric field implies a red shift of the resonant structures of the ACs and RICs, this effect being much more visible in the case of the $V_{4}\left(z\right)$ potential.

## 4. Conclusions

In the present study, we have investigated the absorption coefficients and relative changes in the refractive index for the inter-subband transitions between the lower-lying allowed energy levels in cylindrical quantum dots that have different hyperbolic-type axial potentials. The applied electric field effects have been considered. The results obtained show that the geometric shape of the structure and the applied external electric field are very effective on the electronic and optical properties. To control the energy spectrum in cylindrical quantum dots, systems with two structure parameters such as radius and cylinder height, except for the shape of axial potential, provide an advantage for researchers, and this advantage provides more possibilities to investigate the optical properties of nanostructures. As can be seen from the results obtained, the CQD that has double QW potential in the axial direction is most sensitive to the effects of electric fields, due to a greater spatial extension of the structure in the axial direction. The study of the shape effect of the quantum dots under external fields on the electronic spectrum and optical responses plays an important role in semiconductor physics, because it can simulate the real situation, and greatly modulate and optimize the performance of optoelectronic devices based on low-dimensional heterostructures. Among the findings that have appeared in this research, we can summarize the following. (i) The separation of variables for the radial and axial coordinates is a good approximation when considering heterostructures whose dimensions significantly exceed the effective Bohr radius of the QD material. (ii) In general, for all the axial confinement potentials that have been considered, the energy of the confined states are decreasing functions of the *k*-geometric parameter. (iii) The presence of energy degeneracies gives information on symmetry exchanges between optically excited states, so it is not easy to predict the selection rules for the dipole matrix elements in each of the studied potentials. (iv) The resonant structures of the AC and RIC present red-shift when the *k*-parameter is increased. (v) Finally, in the case of the $V_{2}$ potential, the presence of the applied electric field generates a small blue-shift of the resonant structures of the AC and RIC, a situation that is opposite and ostensibly greater in the case of the $V_{3}$ and $V_{4}$ potentials.

We have to stress that this research is of a theoretical and predictive nature, and that we hope that it will serve as a motivation for experimental groups to develop these types of axis-symmetric physical systems. Additionally, this research could motivate further developments that consider, for example, (i) effects of shallow donor and/or acceptor impurities with unintentional doping along the structure growth direction, (ii) exciton states in single and double QWS, (iii) effects of magnetic fields applied perpendicularly and parallel to the QW growth direction and (iv) other optical properties related to electron, impurity, and exciton states such as electromagnetically induced transparency and second and third harmonic generation. Finally, we want to emphasize that our model allows us to study systems with confinement in two or three dimensions such as core/shell quantum wires and core/shell quantum dots, including interdiffusion effects at the interfaces.

## Figures and Tables

**Figure 1 nanomaterials-12-03367-f001:**
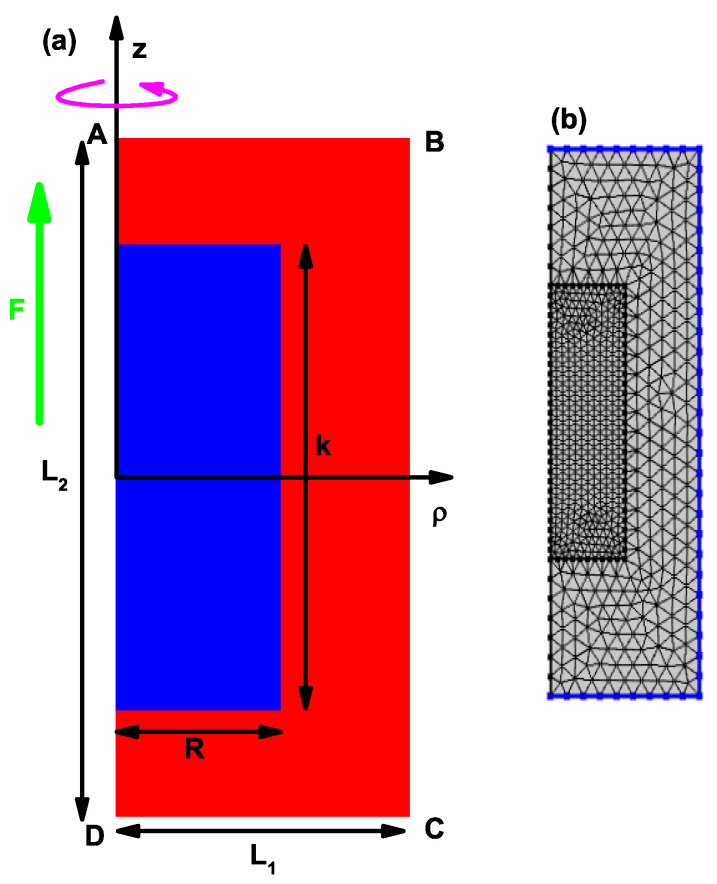
Schematic of the structure used to solve the two-dimensional eigenvalue differential equation, Equation (Equation 3), using the finite element method (**a**). In (**b**), the mesh used with its refinement in the QD region to implement the finite element method is shown. F represents the applied electric field.

**Figure 2 nanomaterials-12-03367-f002:**
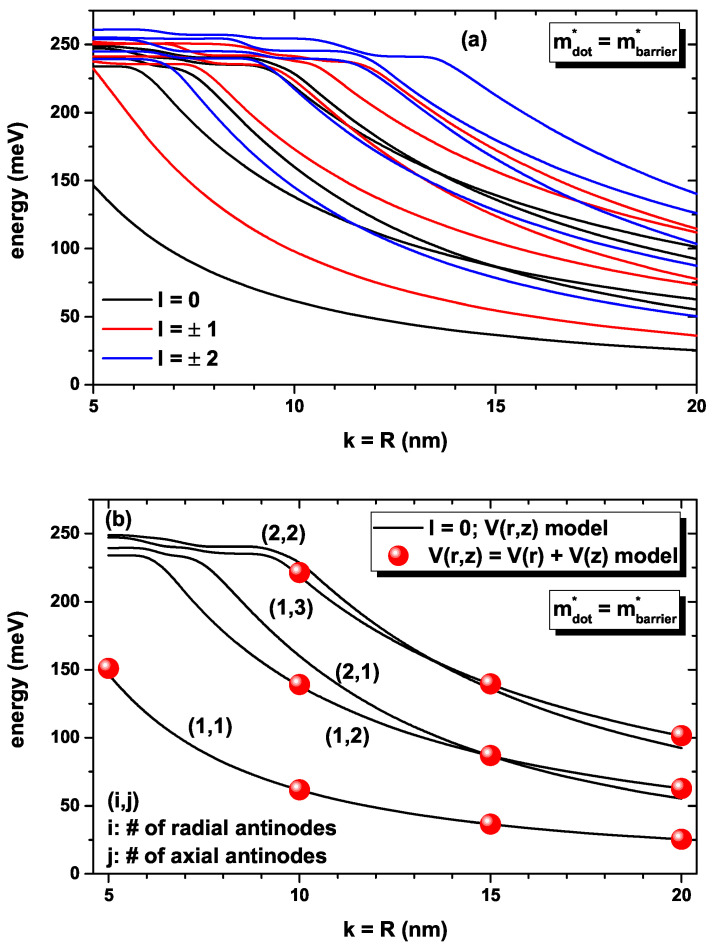
The lowest energies for a confined electron within the CQD as a function of the simultaneous variation of the *k*-width in the axial direction and the *R*-radius in the radial direction. Calculations are for zero applied electric field and using the model $V_{1}\left(z\right)=V_{0}\;\sinh^{2}\left(\frac{z}{k}\right)$ for the axial confinement potential (see Equation (Equation 7)). In (**a**,**b**), the solid lines correspond to the results obtained by the solution of the full non-separable V(ρ,z) confinement potential (this means that they are obtained by solving the 2D-axi-symmetric differential equation). In (**a**), the five lowest solutions are reported for l=0,±1,±2. In (**b**), the full symbols are obtained by the approximation $V\left(\rho ,z\right)=V_{\rho }\left(\rho \right)+V_{1}\left(z\right)$ using only the first solution for the radial differential equation. In (**b**), the (i,j) labels identify the number of wave function antinodes, both in the radial (i) and axial (j) directions.

**Figure 3 nanomaterials-12-03367-f003:**
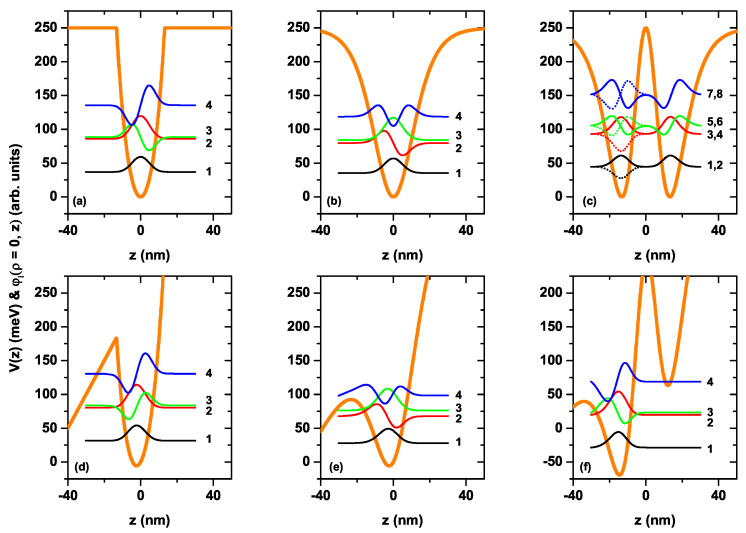
The *z*-dependent axial finite confinement potential ($V_{n}\left(z\right)$) and the wave functions along the *z*-axis of first four bound-states ($\varphi_{i}\left(\rho =0,z\right)$) for a confined electron in a CQD with k=R=15 nm. Results are for zero (**a**–**c**) and 50 kV/cm (**d**–**f**) applied electric field, with l=0. Calculations are reported for the three considered axial finite confinement potentials: $V_{2}\left(z\right)$ (**a**,**d**), $V_{3}\left(z\right)$ (**b**,**e**) and $V_{4}\left(z\right)$ (**c**,**f**). Each wave function is vertically shifted such that its position on the energy scale is given by its corresponding eigenvalue. In panel (**c**), there are four pairs of doubly degenerate states, one with even symmetry (solid line) and the other with odd symmetry (dotted line) with respect to the point z=0. The labels in each figure indicate the states in order of increasing energy.

**Figure 4 nanomaterials-12-03367-f004:**
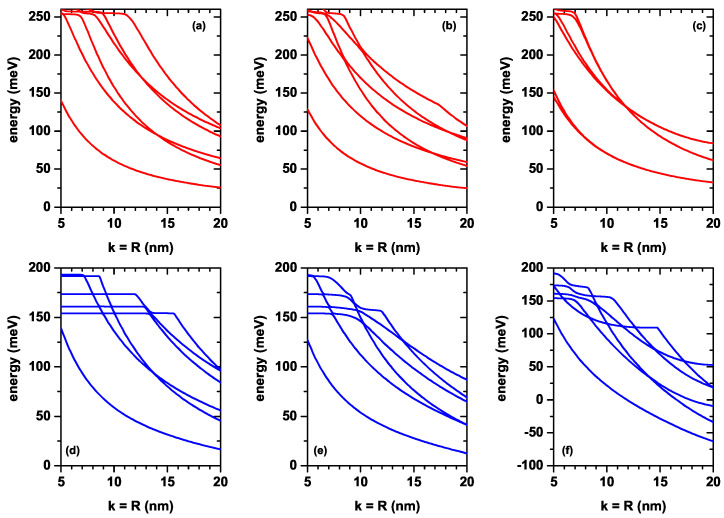
The lowest six energies for a confined electron within the CQD as a function of the simultaneous variation of the *k*-width in the axial direction and the *R*-radius in the radial direction. Calculations are for zero (**a**–**c**) and F=50 kV/cm (**d**–**f**) with l=0. The results are reported for the three considered axial finite confinement potentials: $V_{2}\left(z\right)$ (**a**,**d**), $V_{3}\left(z\right)$ (**b**,**e**) and $V_{4}\left(z\right)$ (**c**,**f**).

**Figure 5 nanomaterials-12-03367-f005:**
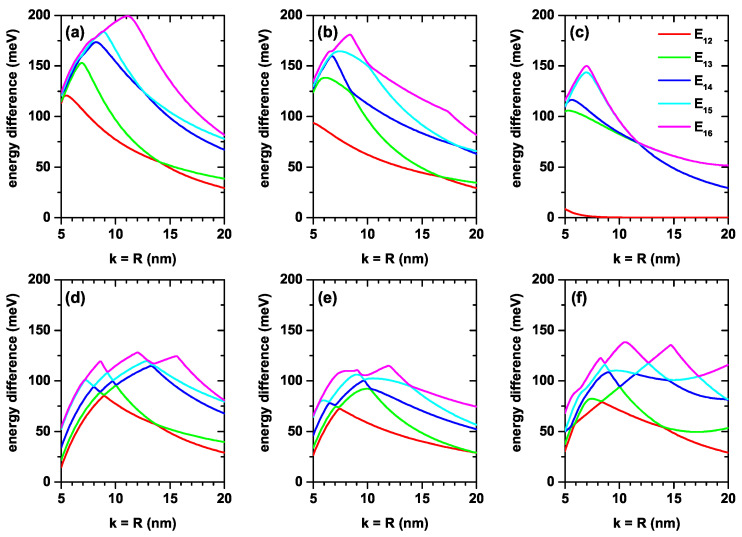
Energy difference between the lowest confined electron states ($E_{n}-E_{1}$, n=2,3,4,5,6) within the CQD as a function of the simultaneous variation of the *k*-width in the axial direction and the *R*-radius in the radial direction. Calculations are for zero (**a**–**c**) and F=50 kV/cm (**d**–**f**) with l=0. The results are reported for the three considered finite axial confinement potentials: $V_{2}\left(z\right)$ (**a**,**d**), $V_{3}\left(z\right)$ (**b**,**e**) and $V_{4}\left(z\right)$ (**c**,**f**).

**Figure 6 nanomaterials-12-03367-f006:**
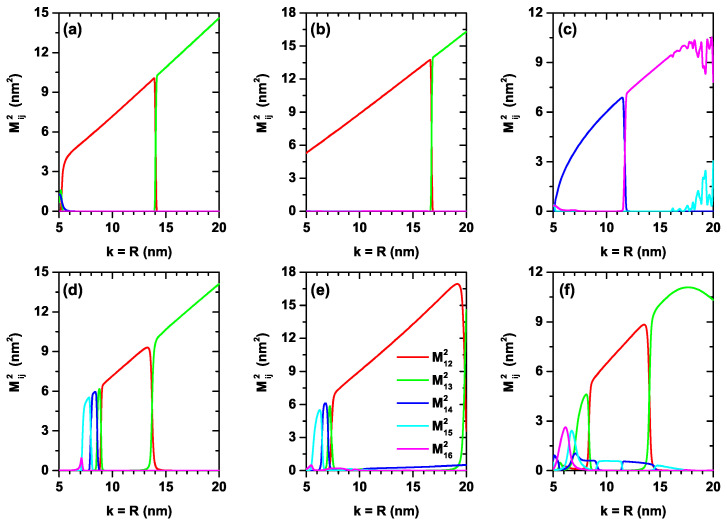
Reduced square of the dipole matrix element between the lowest confined electron states ($M_{1n}^{2}/e^{2}$, n=2,3,4,5,6) within the CQD as a function of the simultaneous variation of the *k*-width in the axial direction and the *R*-radius in the radial direction. Calculations are for zero (**a**–**c**) and F=50 kV/cm (**d**–**f**) with l=0. The results are reported for the three considered finite axial confinement potentials: $V_{2}\left(z\right)$ (**a**,**d**), $V_{3}\left(z\right)$ (**b**,**e**) and $V_{4}\left(z\right)$ (**c**,**f**).

**Figure 7 nanomaterials-12-03367-f007:**
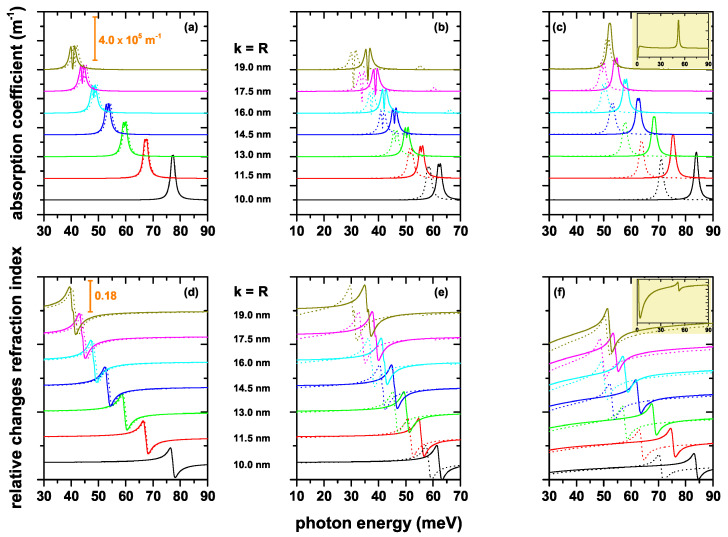
Optical absorption coefficient (**a**–**c**) and relative changes of the refraction index (**d**–**f**) as a function of the incident photon energy for a confined electron within the CQD. Calculations are for several values of the k=R parameters for zero (solid lines) and F=50 kV/cm (dotted lines), considering transitions between l=0 states. The results are reported for the three considered axial finite confinement potentials: $V_{2}\left(z\right)$ (**a**,**d**), $V_{3}\left(z\right)$ (**b**,**e**) and $V_{4}\left(z\right)$ (**c**,**f**). The insets in (**c**,**f**) show the optical coefficients for k=R=19 nm for the energy photon between zero and 90 meV in order to see the transitions between states that are almost degenerated. (**a**,**d**) show the scales for the optical coefficients.

## Data Availability

No new data were created or analyzed in this study. Data sharing is not applicable to this article.

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
