# Peer review of "Optical Properties of Cylindrical Quantum Dots with Hyperbolic-Type Axial Potential under Applied Electric Field"

_nanomaterials, 2022, doi:10.3390/nano12193367_

Round 1

Reviewer 1 Report (Previous Reviewer 1)

I have gone through the response to my comments on their work and the modified text. I believe they have adequately addressed all my concerns.  

I do have a final but minor comment on the modified draft: 

I suggest the authors to not use the words "for the first time" in the abstract. This is because, the way it is written it sounds like they have considered the optical properties spherical quantum dots for first time. Of course, their intended meaning was that they have computed optical/electrical properties under very specific set of circumstances. My suggestion is that they remove the words "for the first time" from the abstract. 

Author Response

The Referee:

I have gone through the response to my comments on their work and the modified text. I believe they have adequately addressed all my concerns.

I do have a final but minor comment on the modified draft:

I suggest the authors to not use the words "for the first time" in the abstract. This is because, the way it is written it sounds like they have considered the optical properties spherical quantum dots for first time. Of course, their intended meaning was that they have computed optical/electrical properties under very specific set of circumstances. My suggestion is that they remove the words "for the first time" from the abstract.

Our reply:

We want to thank the Referee for his/her comments and suggestions and also for his/her positive report. We have removed in the Abstract of the revised version of the manuscript the words "for the first time"

We hope that the revised version of our manuscript will be suitable for publication in the Special Issue Optical Properties of Semiconductor Nanomaterials of the Journal Nanomaterials.

Reviewer 2 Report (Previous Reviewer 2)

can be accepted.

Author Response

We want to thank the Referee for his/her positive answer

This manuscript is a resubmission of an earlier submission. The following is a list of the peer review reports and author responses from that submission.

Round 1

Reviewer 1 Report

Kasapoglu et. al. investigates optical properties and eigenspectra of cylindrical quantum dots with various confinement potential with and without static electric field. The authors find that static electric field distorts the confinement potential leading which modifies the eigenspectra as well as absorption coefficients.  

Overall, the results are not surprising, and a wide range of similar works has been done previously by some of the authors themselves and others [such as Optik 236 (2021) 166621, Photonics and Nanostructures  41, (2020) 100833, Physica B 474, 1 (2015), 15-20 etc.]. In my opinion the work is more suitable for a more specialised journal. In addition to this, I have some comments that the authors can address,

  1. The authors consider four different confinement potentials – is there a physical significance to these potentials ? Is it possible to manufacture materials that will resemble such potential ? If this is done purely as a theoretical exploration the author should mention this in introduction and/or in conclusions.

  2. The authors write the theoretical model without any citations. Similar Hamiltonians have been used extensively- and they should cite previous works that discuss such Hamiltonian and other equations in more detail. Parts of this section (theoretical model) are very confusing too, for example eqn. 11 has been introduced without any details of why x(z) is chosen to take sucha form – where do these numbers such as “1155” come from ? Similarly eqn.12 Why are these specific numbers chosen? Provide citations and/or background to Eqn. 19- 21.

  3. The authors write: “It is clear from Fig. 2(b) that regardless of the values of k = R, the ground state always has one antinode in the radial and axial directions.” Isn't this almost always true for arbitrary potential to have antinodes in the ground state wavefunction (unless it's a free particle)?

  4. With the electric field switched on, does the potentials become unbounded (such as in Fig.3d-f) ? or they are guaranteed to be bounded at large z? 

Reviewer 2 Report

In the abstract section, I do not see the importance of the research. What problem does this work attempt to solve? As for implications of your research, what changes should be implemented as a result of the findings of the work? How does this work add to the body of knowledge on this topic? Contribution of the said study seems marginal and not convincing. Limitations of study are not stated as well.

 Likewise for the introduction section, the research problem remains vague. The authors should strengthen theproblem statement by defining the problem being addressed in a way that is clear and concise.

 What are the limitations or the deficiencies of those past studies that your study purports to solve? Furthermore, the research objective(s) seems unclear. What is/are the general and specific research objectives for this study?

Where is the literature review section in this paper? It is essential to have a literature review section to look for any drawbacks or limitations in their methodology. Comment on why you may be reluctant to trust their conclusions. In terms of applications/conclusion/implications, did your findings suggest a need for further research, what might this consist of and how might such research extend or improve the current state of knowledge in this field? What is your contributions to theory?